Construction and verification of a prognostic model for bladder cancer based on disulfidptosis-related angiogenesis genes

Zhou Zhihao 1 2
Zhang Yuwei 3
Zhou Yuhua 1 2
Gu Jiayu 1 2
Li Jufa 1 2
Shao Jianfeng 1 2 jianfenguro@163.com
Feng Ninghan 1 2 n.feng@jiangnan.edu.cn
1 Wuxi School of Medicine, Jiangnan University , Wuxi , China
2 Department of Urology, Jiangnan University Medical Center , Wuxi , China
3 Medical School of Nantong University , Nantong , China
Nakai Kenta
Electronic publication date: 2025 Feb 21
Publication date: 2025
Volume: 13
Electronic Location ID: e18911
Received 2024 Sep 30; Accepted 2025 Jan 6
Copyright: © 2025 Zhou et al.
Copyright year: 2025
Copyright holder: Zhou et al.
License: This is an open access article distributed under the terms of the Creative Commons Attribution License, which permits unrestricted use, distribution, reproduction and adaptation in any medium and for any purpose provided that it is properly attributed. For attribution, the original author(s), title, publication source (PeerJ) and either DOI or URL of the article must be cited.
License URL: https://creativecommons.org/licenses/by/4.0/

Keywords: Bladder cancer, Disulfidptosis, Angiogenesis, Prognostic model, Immune

Funding: Postgraduate Research & Practice Innovation Program of Jiangsu Province KYCX24_3596 The study was funded by the Postgraduate Research & Practice Innovation Program of Jiangsu Province (Item No: KYCX24_3596). The funders had no role in study design, data collection and analysis, decision to publish, or preparation of the manuscript.

==============================
Background

Bladder cancer (BLCA) is the most common malignancy of the urinary system and one of the most common cancers worldwide. This study seeks to examine the influence of angiogenesis-related genes (ARGs) linked to disulfidptosis on BLCA patients and to formulate a prognostic model for evaluating their prognosis and response to immunotherapy.

Methods

This study used sequencing data of BLCA in the Cancer Genome Atlas (TCGA) database. Unsupervised consensus clustering analysis, cox regression analysis, and least absolute shrinkage and selection operator (LASSO) regression analysis were used to screen hub genes and construct a related prognostic risk model. The receiver operating characteristic (ROC) curve and independent prognostic analysis were then used to verify the predictive performance of the signature genes. Clinical characteristics, immune status, and Tumor Mutation Burden (TMB) of the prognostic risk model were evaluated. The expression levels of model genes within standard bladder epithelial cell lines (SV-HUC-1) and bladder cancer cell lines (T24 and SW1710) were quantified through qRT-PCR.

Results

The constructed prognostic risk model can be used as an independent risk indicator for BLCA and was validated in an external dataset. Immune cell infiltration analysis showed that CD8+T cells, Tregs and dendritic cells were significantly different between the two groups. A significant increase was observed in the Stromal score, Immune score and ESTIMATE score in the high-risk group compared with the low-risk group. The Immune Exclusion score and Tumor Immune Dysfunction and Exclusion (TIDE) score of the high-risk group were higher than those of the low-risk score group. Compared with the normal bladder epithelial cell line (SV-HUC-1), the expression levels of 2 model genes (COL5A2 and SCG2) in bladder cancer cell lines (T24 and SW1710) were significantly elevated.

Conclusion

This study helps us understand the characteristics of disulfidptosis-related subgroups. The characteristics of disulfidptosis-related ARGs may be used to evaluate the prognosis and immunotherapy response of BLCA patients.

Introduction

Bladder cancer (BLCA) represents the most frequent malignant condition of the urinary tract. It has an annual incidence of approximately 550,000 new cases worldwide, with around 200,000 deaths each year. The incidence among males is markedly higher than that of females, with males accounting for approximately 75% of newly diagnosed cases (Contieri et al., 2023; Kamat et al., 2016; Wang et al., 2022). This medical condition is fundamentally classified into two subtypes: non-muscle-invasive BLCA (NMIBC) and muscle-invasive BLCA (MIBC) (Cao et al., 2020). Upon initial diagnosis, 75% of patients are found to have NMIBC, whereas the remaining 25% are diagnosed with MIBC (Tan et al., 2019). Despite significant improvements in the prognosis of NMIBC patients due to advances in medical technology, more than 20% of high-risk NMIBC cases advance to MIBC (Lenis et al., 2020). Patients diagnosed with MIBC generally have a worse prognosis. Since cancer is a genetic disease driven by oncogenic alterations, treatments targeting the genes with oncogenic mutations and related signaling pathways remain an important approach in cancer therapy (Sonkin, Thomas & Teicher, 2024). Therefore, identifying specific biological markers for BLCA screening and exploring effective molecular therapeutic targets is imperative.

Cell death can be categorized into regulated cell death (RCD) and accidental cell death (ACD). ACD occurs passively due to external factors such as physical or chemical damage, without the clear involvement of internal signals or execution mechanisms. Contrary to ACD, RCD is managed by specific signaling pathways, and it can be regulated by pharmacological or genetic approaches (Tong et al., 2022). RCD encompasses types such as necroptosis, pyroptosis, ferroptosis, and cuproptosis. They all possess distinct molecular mechanisms. For instance, ferroptosis is triggered by the accumulation of iron-dependent lipid peroxides (Garciaz et al., 2023), pyroptosis is initiated by specific signaling pathways involved in inflammation (Wang et al., 2017), and cuproptosis is caused by the toxic stress from acylation of tricarboxylic acid cycle proteins in the mitochondria (Liu, 2022, 2023). In recent years, Liu et al. (2023) have unveiled a novel mode of cellular demise, namely disulfidptosis. Their research revealed that under glucose starvation, disulfides, including intracellular cystine, aberrantly amass in cells with raised expression levels of SLC7A11, inducing disulfide stress and enhancing the level of disulfide bonds within the actin cytoskeleton. As a consequence, actin filaments contract, disrupting cytoskeletal structure, and ultimately leading to cellular demise. This manner of cell death differs from established modes such as apoptosis and ferroptosis (Liu et al., 2023). According to a preclinical study, metabolic therapies utilizing glucose transporter inhibitors can induce disulfidptosis and reduce cancer growth (Xiao et al., 2024). Certain drugs can induce or inhibit disulfidptosis by modulating the expression of SLC7A11 or altering the concentration of disulfide bonds, thereby suppressing cancer growth and metastasis (Zhao et al., 2023a). This creates more pathways for finding targeted approaches to cancer therapy. Nevertheless, the prognostic significance of disulfidptosis-related genes (DRGs) in BLCA and their underlying regulatory mechanisms remain to be elucidated.

Angiogenesis, the formation of novel blood vessels originating from already established vessels, can be categorized into two types: normal physiological angiogenesis and pathological angiogenesis. Normal angiogenesis is essential for the formation and growth of vascular systems (Griffioen & Dudley, 2022). Nevertheless, pathological angiogenesis is a distinguishing trait of cancer and has been demonstrated to be a fundamental prerequisite for the sustained growth and proliferation of tumors. This pathological process is supported by an array of growth-promoting factors, cytokines, peptides, and enzymes, either individually or in concert (Elayat, Punev & Selim, 2023; Lee & Seo, 2021). Nevertheless, a reliable predictive model based on genes linked to angiogenesis for individuals with BLCA has yet to be established.

This study drew upon datasets from The Cancer Genome Atlas (TCGA) and the Gene Expression Omnibus (GEO) databases to undertake an unsupervised consensus clustering analysis. BLCA patients were categorized into two disulfidptosis-related subgroups. Subsequently, differential analysis, in conjunction with enrichment analyses, was carried out on disulfidptosis-related differentially expressed genes (DEGs). Subsequently, 82 angiogenesis-related genes (ARGs) were earmarked and then intersected with the DEGs. Common genes were then singled out for least absolute shrinkage and selection operator (LASSO) regression analysis. Ultimately, two risk genes were determined and subsequently leveraged to develop a prognostic model. To validate the efficiency of this model in forecasting the prognosis of BLCA, receiver operating characteristic (ROC) curves and survival graphs were created, and the GEO dataset (GSE32548) was leveraged. Furthermore, pertinent bioinformatics algorithms were utilized to ascertain the interplay between risk scores and clinical features, immune status, as well as tumor mutation burden. Finally, the model was validated using qRT-PCR. We formulated and verified a prognostic model built upon ARGs associated with disulfidptosis, which was highly precise in forecasting the prognosis and immune response across different independent cohorts. The findings from this bioinformatics analysis present fresh insights into the genetic mechanisms underlying the development of BLCA. The identified key risk genes may offer promising targets for therapeutic interventions.

Materials and Methods

Data gathering and processing

The study workflow is presented in Fig. 1A. An aggregate of 425 BLCA samples were procured from the TCGA repository (https://portal.gdc.cancer.gov/), comprising 406 tumor samples as well as 19 normal specimens. Among the 406 BLCA patients, 401 had complete survival data available for analysis (Table S1). Additionally, the GSE32548 cohort, comprising 131 BLCA patients, was obtained from the GEO repository (https://www.ncbi.nlm.nih.gov/geo/query/acc.cgi?acc=GSE32548) and used for external verification (Table S2). DRGs were sourced from a presently available publication (Xin et al., 2023). The 82 ARGs were retrieved from the GESA Molecular Signatures Database (https://www.gsea-msigdb.org/gsea/msigdb/cards/HALLMARK_ANGIOGENESIS) (Subramanian et al., 2005).

Figure 1 Study process flowchart.

Unsupervised clustering analysis

Based on 11 DRGs, an unsupervised clustering analysis was executed on the 406 tumor specimens from the TCGA-BLCA cohort by applying the ConsensuClusterPlus software package (Wilkerson & Hayes, 2010). This analysis aimed to identify DRG-related subgroups. The key clinical information among the subgroups was organized and compared (Table S3). The “Survival” and “survminer” packages (version 4.22) were employed to generate Kaplan-Meier (KM) curves of OS, thereby determining the prognostic differences between the identified subgroups. P < 0.05 was indicative of statistical significance.

Differential analysis and enrichment analysis

To elucidate the mechanisms of DRGs, the “DESeq2” software package was applied to carry out differential expression analysis (Love, Huber & Anders, 2014). Genes exhibiting differential expression were identified in line with the criteria of P < 0.05, |log2FC|≥ 1. Following this step, the “clusterProfiler” package was leveraged to execute Gene Ontology (GO) and Kyoto Encyclopedia of Genes and Genomes (KEGG) enrichment analyses (Yu et al., 2012), with identical filtering criteria. Furthermore, the “VennDiagram” software package was utilized to intersect disulfidptosis-related DEGs with the selected 82 ARGs.

Formulation and verification of a model based on disulfidptosis-related ARGs

Through LASSO regression analysis, two key genes were identified. The risk score was derived via the formula: Risk Score = Σ (Expi × Coefi), in which Coefi denotes the risk coefficient, while Expi signifies the expression level of each gene. BLCA samples were then segmented into low-risk and high-risk cohorts leveraging the median risk score. Prognostic analysis, along with ROC curve analysis, was carried out via “ggrisk” “survival” and “rms” software packages. In the test cohort, the robustness of the model was further affirmed.

Clinical relevance analysis

To ascertain the influence of BLCA-linked variables on the outcome of prognosis, both univariate and multivariate Cox regression analyses were implemented. Subsequently, we probed the linkage between the prognostic risk model and various clinical features, covering age, gender, clinical stage, as well as pathological TNM stage, as obtained from TCGA. To forecast BLCA patients’ survival probability, the “rms” and “survival” packages were applied to formulate a nomogram. For each patient, a total score was derived from the values assigned to the predictive factors, and this total score was thereafter used to project the likelihood of survival (Park, 2018). The efficiency of the constructed nomogram was evaluated by calibration curves.

Immunological analysis and tumor mutation burden

The “CIBERSORT” package was employed to quantify immune cell’s degree of infiltration in both high- and low-risk cohorts (Newman et al., 2015). Differences in immune checkpoint expression between the high- and low-risk cohorts were compared via “tinyarray” software package. Subsequently, the R package “estimate” was leveraged to estimate and compare Stromal scores, Immune scores, and ESTIMATE scores across the two cohorts at risk. Online tool TIDE (https://tide.nki.nl/) was then implemented to assess the potential for evasion of immune response. The TCGA database provided somatic mutation information for BLCA patients, while waterfall plots illustrating the gene mutation distribution between the high- and low-risk segments were generated via R package “maftools.” Survival curves integrating risk scores with tumor mutation burden (TMB) were constructed and analyzed to gauge the prognosis of BLCA patients.

Cell cultivation and qRT-PCR

The standard bladder epithelial cell strain (SV-HUC-1) and the bladder cancer cell strain (T24 and SW1710) were sourced from the Cell Bank of the Chinese Academy of Sciences (Shanghai). These cell strains were cultured in RPMI 1640 medium (Gibco, Waltham, MA, USA), enriched by 10% fetal bovine serum (Gibco, Waltham, MA, USA) and 1% penicillin/streptomycin (Gibco, Waltham, MA, USA), and incubated at 37 °C in 5% CO2. Total RNA was isolated using TRIzol reagent (Invitrogen, Waltham, MA, USA). Reverse transcription was performed using HiScript III SuperMix (Vazyme, Beijing, China). qPCR assays were carried out with a QuantStudio 3 instrument (Thermo, China). Briefly, Upon the denaturation at 95 °C for 1 min, these cell strains were amplified at 95 °C for 5 s and 60 °C for 15 s for 40 cycles. The mRNA expression levels of COL5A2 and SCG2 were quantified through the 2−ΔΔCt approach. The primers utilized in these reactions were presented here:

β-ACTIN forward primer: CGGGAAATCGTGCGTGAC; reverse primer: CAGGCAGCTCGTAGCTCTT

COL5A2 forward primer: GGTCTCAGTTCGCTTATGG; reverse primer: TGTAAGTGATGTTCTGGGAGG

SCG2 forward primer: ACCAGACCTCAGGTTGGAAAA; reverse primer: AAGTGGCTTTCATCGCCATTT

Statistical analysis

R software (version 4.4.1) and GraphPad Prism (version 9.2.0) were employed to execute all statistical analyses. The Spearman correlation coefficient was utilized to determine the correlation between two variables. Continuous variables were subject to the Student’s t-test or the Mann-Whitney U test, while the chi-square test was adopted to evaluate categorical variables. Differences were deemed to be statistically significant at P < 0.05.

Results

Identification of disulfidptosis-related subgroups

In the present study, a comprehensive literature review was conducted to identify 24 genes linked to disulfidptosis. Among these 24 DRGs, 11 exhibited variations in the expression levels between BLCA tumor specimens and non-tumorous samples. Specifically, an elevation in the expression levels of PRN1, OXSM, SLC3A2, and CD2AP were observed in tumor tissues, while the expression levels of DSTN, FLNA, TLN1, IQGAP1, MYL6, NCKAP1, and NDUFS1 declined in tumor tissues (Fig. 2A). An unsupervised consensus clustering analysis was carried out on the 11 differentially expressed DRGs to ascertain the biological functions of DRGs in BLCA. Results from the analysis revealed that the TCGA-BLCA cohort could be split into two subgroups when k = 2 (Figs. 2B–2D). A notable disparity in the survival rates between the two subgroups was noted, as indicated by Kaplan-Meier survival analysis (Fig. 2E).

Figure 2 Expression levels of disulfidptosis-related genes and identification of disulfidptosis-related subgroups.

(A) Differential expression of disulfidptosis-related genes between tumor and normal tissues. ns: no significance, **P < 0.01, ***P < 0.001, ****P < 0.0001. (B) Consensus matrix for the BLCA cohort, k = 2. (C) Cumulative distribution function curve for consensus clustering. (D) Delta area plot. (E) KM curves for the two subgroups.

Differential analysis and functional enrichment of DEGs

Differential analysis was conducted between the two identified subgroups, and 2,166 DEGs were determined. Among these DEGs, 682 genes were downregulated, while 1,484 were upregulated (Fig. 3A). To elucidate the potential molecular mechanisms of these DEGs, GO and KEGG pathway analyses were implemented. As a result, the GO analysis for biological processes (BP) indicated that those DEGs were predominantly enriched in such processes as cell chemotaxis, arrangement of the extracellular matrix, organization of extracellular structures, and formation of external encapsulating structures. The GO analysis for cellular components (CC) indicated that these genes were predominantly enriched in the collagen-laden extracellular matrix, the outer side of the plasma membrane, and the lumen within the endoplasmic reticulum. With respect to molecular function (MF), the GO analysis demonstrated that the DEGs were principally involved in the glycosaminoglycan interaction, the construction of extracellular matrix structure, as well as endopeptidase activity (Fig. 3B). Prior research has demonstrated the pivotal role of such processes as the arrangement of the extracellular matrix, organization of extracellular structures, collagen-laden extracellular matrix, and the construction of extracellular matrix structure in regulating tumor angiogenesis (Mongiat et al., 2016; Pickup, Mouw & Weaver, 2014; Zhao et al., 2023b). KEGG pathway analysis uncovered substantial enrichment of DEGs within certain pathways, for instance, cytokine-cytokine receptor engagement, the PI3K-Akt signaling route, neuroactive ligand-receptor interplay, as well as the cytoskeleton of muscle cells (Fig. 3C). According to GO and KEGG analyses, 82 ARGs were selected from the molecular signature database and were subsequently intersected with the 2,166 DEGs. Ultimately, 11 common genes were determined, including COL3A1, COL5A2, EPGN, POSTN, SCG2, SERPINA5, SPHK1, SPP1, STAB1, TGFB2, SHH (Fig. 3D).

Figure 3 Differential analysis and enrichment analysis of disulfidptosis-related gene subgroups.

(A) Volcano plot displaying differentially expressed genes, |log2FC| ≥ 1, p < 0.05. (B) GO analysis of differentially expressed genes. CC: Cellular Component, BP: Biological Process, MF: Molecular Function. (C) KEGG pathway analysis of differentially expressed genes. (D) Venn diagram showing the intersection of differentially expressed genes and disulfidptosis-related genes.

Formulation and verification of prognostic risk model

To ascertain the implications of the 11 disulfidptosis-related ARGs for prognosis, the LASSO algorithm was employed. Two optimal candidate genes (SCG2 and COL5A2) with the minimum lambda were determined and then were subsequently utilized to formulate the risk model (Figs. 4A and 4B). The formula was established as: Risk Score = COL5A2 * 0.12456 + SCG2 * 0.19130. Patients were stratified into high- and low-risk cohorts according to a specific cut-off of −0.04. Kaplan-Meier (KM) survival analysis indicated that high-risk individuals demonstrated a markedly less favorable prognosis when contrasted with low-risk individuals (Fig. 4C). The ROC curve analysis corroborated the prognostic validity of this model, with AUC readings standing at 0.613 for 3 years and 0.624 for 5 years (Fig. 4D). Scatter plots demonstrated that the high-risk population experienced a heightened mortality rate in comparison to the low-risk population, while heatmaps showed elevated expression levels of the risk-related candidate genes (COL5A2 and SCG2) in the high-risk group (Fig. 4E). To further affirm the model’s robustness, the GEO dataset (GSE32548) was selected for external validation. The results confirmed that patients who had higher risk scores had reduced overall survival (OS) and increased fatality (Fig. 4F). At the 3-year and 5-year time points, the AUC values were 0.803 and 0.762, respectively (Fig. 4G), thereby substantiating the stable predictive efficiency of the risk model.

Figure 4 Establishment and validation of the prognostic risk model.

(A) Cross-validation in LASSO regression. (B) LASSO regression analysis. (C) KM curves for high- and low-risk groups in the TCGA cohort. (D) ROC curves for 3-year and 5-year risk scores in the TCGA training cohort. (E) Risk score plot, risk status scatter plot, and heatmap of risk gene expression in the TCGA training cohort. (F) KM curves for high-risk and low-risk groups in the GSE32548 cohort. (G) ROC curves for 3-year and 5-year risk scores in the GSE32548 validation cohort.

Association between prognostic risk model and patient clinical features

To elucidate the interrelation between the risk model and the prognosis of BLCA patients, COX regression analysis and univariate and multivariate analyses were implemented on clinical features and risk scores. The results demonstrated that riskScore, Age, and Stage emerged as independent prognostic indicators for BLCA patients (Figs. 5A and 5B). Utilizing the aforementioned risk score along with clinical-pathological elements, specifically, age, gender, as well as disease progression stage, a nomogram was developed for the purpose of determining probabilities of survival at 1, 3, and 5 years (Fig. 5C). The nomogram’s predictive reliability for prognoses at 1, 3, and 5 years was substantiated by calibration graphs (Fig. S1). Furthermore, the linkage between this risk model and BLCA patients’ clinical features within the TCGA cohort was examined. Meanwhile, violin plots revealed that elevated risk scores were noted in older individuals (Fig. 5D), females (Fig. 5E), individuals with higher stages (Fig. 5F), and those with higher pathological TNM (Tumor Node Metastasis) stages (Figs. 5G–5I). In summary, the constructed prognostic risk model exhibited robust performance in projecting the prognosis of BLCA patients.

Figure 5 Correlation analysis between prognostic risk model and clinicopathological characteristics.

Univariate and multivariate COX regression analysis of risk score, age, gender, stage, and T. (A and B). (C) Nomogram constructed in the TCGA cohort. Violin plots showing differences in risk scores with respect to age (D), gender (E), stage (F), pathological T (G), pathological N (H), and pathological M (I).

Association of risk prediction model with immune checkpoints, and tumor 3.5.1 microenvironment

The “CIBERSORT” and “ESTIMATE” algorithms were leveraged to probe into the interplay between this risk prediction model and the tumor microenvironment. It was noteworthy that substantial disparities in immune cell infiltration were detected between these two risk cohorts. The high-risk population manifested a notable decline in several anti-tumor immune cells (Figs. 6A and 6B). Specifically, the proportions of CD8+ T cells, Tregs, as well as activated dendritic cells decreased in the high-risk cohort. The evidence underpinned that tumors in the high-risk population likely possessed an immune-inhibiting microenvironment, impairing tumor-counteracting immune function and consequently promoting tumor progression. The correlation analysis demonstrated that dendritic cells activated, dendritic cells resting and T cells CD8 exhibited an inverse relationship with COL5A2, SCG2, and riskScore (Fig. 6C). This finding aligned with the previously mentioned results of the immune cell infiltration analysis. Between high- and low-risk populations, 16 immune checkpoint genes with significant differential expression were determined, including CD200, LAIR1, and HAVCR2. All 16 genes demonstrated elevated expression levels in the high-risk cohort (Fig. 6D). Moreover, our study indicated that significantly greater stromal, immune, and ESTIMATE scores were found in the high-risk population as opposed to the low-risk population (Fig. 6E). To ascertain the interplay between risk score and TME, TIDE was employed for the purpose of gauging BLCA immunological features in the high- and low-risk cohorts. Both the exclusion score and TIDE score were elevated in the high-risk population (Figs. 6F and 6G), suggesting that BLCA with high-risk scores exhibited enhanced immune escape and potential non-responsiveness to immune checkpoint inhibitors.

Figure 6 Analysis of immune level differences between high-risk and low-risk groups.

(A) Relative abundance of different immune cell types calculated using the CIBERSORT algorithm. (B) Differences in the relative abundance of various immune cell types between high-risk and low-risk groups. (1: high; 2: low; 3: B cells memory; 4: B cells naïve; 5: Dendritic cells activated; 6: Dendritic cells resting; 7: Eosinophils; 8: Macrophages M0; 9: Macrophages M1; 10: Macrophages M2; 11: Mast cells activated; 12: Mast cells resting; 13: Monocytes; 14: Neutrophils; 15: NK cells activated; 16: NK cells resting; 17: Plasma cells; 18: T cells CD4 memory activated; 19: T cells CD4 memory resting; 20: T cells CD4 naïve; 21: T cells CD8; 22: T cells follicular helper; 23: T cells gamma delta; 24: T cells regulatory (Tregs)). (C) Correlation of risk score and risk genes with the relative abundance of different immune cell types. (D) Differences in immune checkpoint expression between high-risk and low-risk groups. (E) TME scores, including Stromal score, Immune score, and ESTIMATE score. (F) TIDE scores in high-risk and low-risk groups. (G) Exclusion scores in high-risk and low-risk groups. ns: no significance, *P < 0.05, **P < 0.01, ***P < 0.001, ****P < 0.0001.

Linkage between prognostic risk model and TMB

In an attempt to delve into the somatic mutation profiles between the two risk cohorts, TMB was analyzed. The results demonstrated that the mutation rate among the 198 high-risk patients stood at 93.94% (Fig. 7A), while it was 96.97% for the 198 low-risk patients (Fig. 7B). The frequency of TMB appeared to be reduced in the high-risk population as opposed to the low-risk population. The interplay between TMB and OS among individuals with BLCA was examined. Our findings indicated that patients possessing H-TMB experienced a superior prognosis compared to those with L-TMB (Fig. 7C). Furthermore, patients were subdivided into four groups according to TMB and risk scores. The findings demonstrated that the L-TMB + high-risk group exhibited the poorest prognosis among the four groups (P < 0.0001) (Fig. 7D). This analysis offers insights into the subtle differences between the genetic profiles of BLCA and their associated prognosis.

Figure 7 Mutation analysis based on prognostic risk model.

(A and B) Waterfall plots summarizing mutations in the high-risk and low-risk patient groups. (C) Kaplan-Meier curves for high and low TMB groups. (D) Kaplan-Meier curves for four groups classified by risk score and tumor mutation burden.

In vitro validation of two gene expressions

To ascertain the credibility of the two genes incorporated in the risk prediction model, we conducted quantitative real-time reverse transcription polymerase chain reaction (qRT-PCR) with standard bladder epithelial cell strain (SV-HUC-1) and bladder cancer cell strains (T24 and SW1710). As illustrated in Figs. 8A and 8B, mRNA expression levels of COL5A2 and SCG2 were considerably elevated in the bladder cancer cell strains (T24 and SW1710) relative to standard bladder epithelial cell strain (SV-HUC-1). The evidence corroborated that the two risk genes could function as diagnostic biomarkers for BLCA.

Figure 8 qRT-PCR analysis of COL5A2 (A) and SCG2 (B) mRNA Levels in Human Normal Bladder Epithelial Cell Line (SV-HUC-1) and Bladder Cancer Cell Lines (T24 and SW1710).

**P < 0.01, ****P < 0.0001.

Discussion

BLCA is acknowledged to be the second most widespread malignant condition within the genitourinary tract (de Braud et al., 2002). Recent advancements in treatment strategies for patients with advanced-stage disease have introduced immune checkpoint inhibitor therapy, precision-targeted therapies, and antibody-drug combinations as promising interventions across various stages of the illness (Lenis et al., 2020). Nevertheless, the prognosis of individuals with advanced BLCA remains unfavorable. Disulfidptosis, a newly identified form of cell death, is distinguished by the accumulation of disulfide bonds, which gives rise to cytoskeletal collapse and subsequent cell death (Liu, Zhuang & Gan, 2024). Nonetheless, the specific mechanisms of disulfidptosis, along with its regulatory roles and potential pathways in diverse diseases, remain insufficiently explored. Accordingly, this study constructed a predictive model derived from disulfidptosis-related DEGs. Such a model may unveil fresh approaches to treating BLCA.

Using the TCGA-BLCA dataset, unsupervised clustering analysis was performed on DRGs, and subsequently, the samples were categorized into two groups prior to a differential analysis. The enrichment analysis highlighted that DEGs were mostly concentrated in pathways concerning angiogenesis. By means of LASSO regression analysis, two genes associated with angiogenesis were determined, which were then harnessed to formulate a prognostic model. For the purpose of evaluating BLCA prognosis and guiding therapeutic strategies, we examined the linkage between the risk score and a range of clinical traits, including CIBERSORT, immune status, TIDE, and tumor mutation burden. ROC graphs and survival graphs were formulated and validated in the GEO dataset. This study is the first to formulate a risk prediction model via intersecting disulfidptosis-related DEGs with ARGs, which is never reported in previous publications.

SCG2, a critical member of the granule protein family, regulated by AP-1, is a neuroendocrine protein crucial for the formation and biosynthesis of secretory granules (Fang et al., 2021). SCG2 has been increasingly demonstrated to be highly expressed in colorectal carcinoma, renal carcinoma, melanoma, as well as other cancers (Fukumoto et al., 2023; Steinfass et al., 2023; Weng et al., 2022). In colorectal cancer (CRC), a study has identified that SCG2 is linked to the infiltration of tumor immune cells, promotes M2 macrophage polarization, and correlates with the expression of immune checkpoints in CRC (Fukumoto et al., 2023). In BLCA, studies have demonstrated that SCG2 promotes expansion, dissemination, and penetration of BLCA via activating MEK/Erk and MEK/IKK/NF-κB signaling routes and facilitating M2 macrophage polarization (Zhou et al., 2024). These studies imply that SCG2 may be crucial in the tumor immune microenvironment. COL5A2, classified within the type V collagen family, encodes a fibrous collagen alpha chain with low abundance (Cortini & Villa, 2018). Overexpression of COL5A2 has been detected in patients with gastric, prostatic, and colorectal malignancies, and thus can predict the prognosis of tumor patients (Januchowski et al., 2014; Ren et al., 2021; Tan et al., 2021). When COL5A2 is overexpressed, there is an increase in the expression of associated cytokines, including P53 and VEGF, leading to uncontrolled tumor cell growth and angiogenesis (Ding, Sun & Zhao, 2021). COL5A2 modulates tumor development, progression, and treatment response by affecting various components of the tumor microenvironment, such as the extracellular matrix, immune cell function, and angiogenesis. It is essential in the interaction between tumor cells and the matrix, and may represent a new therapeutic target for cancer (Meng et al., 2018). This study unraveled that individuals in the high-risk cohort had augmented SCG2 and COL5A2 mRNA expression. Validation of the in vitro cell model confirmed that these two risk genes manifested heightened expression levels in BLCA cell strains when set against standard bladder epithelial cell strains, corroborating previous research findings. Additionally, a comprehensive analysis of the clinical characteristics of BLCA patients was executed in accordance with the prognostic risk model. The findings revealed that older age, advanced stage, and higher TNM classification were correlated with increased risk scores. This finding unravels that the risk model has reliable efficiency in forecasting the prognosis for BLCA patients. In conclusion, these two risk genes can serve as valuable predictors of tumor prognosis and treatment guidance.

TME constitutes a critical element in tumor pathogenesis (Vinay et al., 2015). Tregs are frequently described as a “double-edged sword” within the human body. It has been demonstrated that a deficiency or impaired function of Tregs can result in an overactive immune response, which may in turn lead to the development of autoimmune diseases. Conversely, an excessive number or aberrant activation of Tregs can suppress immune responses, which is closely linked to the occurrence of various tumors (Scott et al., 2021). In BLCA, individuals with higher infiltration levels of Tregs demonstrate improved prognoses (Winerdal et al., 2018). In general, increased CD8+ T cell infiltration is often indicative of favorable survival for patients (van der Leun, Thommen & Schumacher, 2020). Dendritic cells, as the most efficacious antigen-presenting cells of the immune system, can trigger the activation of naive T cells. Dendritic cells are instrumental in the capture, processing, and presentation of tumor antigens, and they thereby play a pivotal role in the onset, regulation, and sustenance of antitumor immune responses (Calmeiro et al., 2020; Sabado, Balan & Bhardwaj, 2017). This study revealed an elevation in the quantities of Tregs, CD8+ T and dendritic cells in the low-risk group relative to the high-risk group, aligning with prior research findings. The analysis examining the interplay between the risk rating and expression profiles of immune checkpoint genes revealed a positive link between the risk rating and the expression of 16 immune checkpoint genes. This suggests that BLCA patients in the high-risk-score cohort could display a heightened responsiveness to immune checkpoint inhibitors. The TIDE method allows for the evaluation of the likelihood of cancer immune escape based on the genetic expression patterns of malignant samples. Elevated TIDE and Exclusion score signify a greater likelihood of immune evasion (Jiang et al., 2018). The present study yielded analogous findings. The analysis of immunological features in high-risk and low-risk cohorts unveiled that the high-risk cohort demonstrated increased Exclusion and TIDE scores relative to the low-risk cohort. TMB is typically characterized by the aggregate number of mutations in a tumor sample. Patients with higher TMB generally exhibited improved survival rates and enhanced response to immunotherapy (Marabelle et al., 2020). The analysis of the frequency of somatic mutations unraveled a higher mutation frequency in the low-risk cohort. Furthermore, individuals with high TMB within the low-risk cohort exhibited the most auspicious prognoses, thereby further suggesting the predictive performance of this risk model.

The present study has certain limitations. The data obtained from public databases may carry some bias, as the samples are mostly from Caucasians, which could limit the applicability of the results to other ethnic groups. Although both internal and external validations are conducted, there is still a need for larger, racially diverse clinical samples and external datasets for further validation to determine their clinical applicability. Moreover, the specific mechanisms of key genes in BLCA remain unclear, and more in vivo and in vitro experiments are needed to explore the biological roles of these key genes in BLCA. In future studies, we plan to increase the sample size and incorporate technologies such as proteomics and machine learning to further explore the diagnostic performance of this prognostic model (Liu, Guo & Wang, 2024). Cell and animal models will also be employed to validate the function of key genes in the model in relevant signaling pathways related to tumor development and treatment efficacy.

In conclusion, this study employed bioinformatic analysis of the DRGs to formulate a predictive model for BLCA on the foundation of disulfidptosis-related ARGs. Unlike existing BLCA prognostic models, this model is the first to combine genes associated with disulfidptosis and angiogenesis. It demonstrates high accuracy in predicting prognosis and immune response across different independent cohorts. In the future, risk scores could be used to tailor immunotherapy and combination treatments in clinical settings, while validating the model in prospective cohorts and exploring the mechanisms of COL5A2 and SCG2 in BLCA progression. This bioinformatics exploration unveils fresh insights into the development of BLCA and the genetic mechanisms involved in its progression. The key genes have the potential to function as therapeutic targets.

Supplemental Information

Supplemental Information 1 Clinical information of the TCGA cohort.

Supplemental Information 2 Clinical information of the GEO cohort.

Supplemental Information 3 The calibration plot of the nomogram (A-C).

Supplemental Information 4 CONSORT checklist.

Supplemental Information 5 Raw data exported from the Fluorescent Quantitative PCR Instrument applied for data analyses and preparation for Picture A in Figure 8.

Supplemental Information 6 Raw data exported from the Fluorescent Quantitative PCR Instrument applied for data analyses and preparation for Picture B in Figure 8.

Supplemental Information 7 MIQE checklist.

Supplemental Information 8 Clinical information of two subgroups.

Additional Information and Declarations

Competing Interests

The authors declare that they have no competing interests.

Author Contributions

Zhihao Zhou conceived and designed the experiments, authored or reviewed drafts of the article, and approved the final draft.

Yuwei Zhang conceived and designed the experiments, authored or reviewed drafts of the article, and approved the final draft.

Yuhua Zhou performed the experiments, authored or reviewed drafts of the article, and approved the final draft.

Jiayu Gu performed the experiments, analyzed the data, authored or reviewed drafts of the article, and approved the final draft.

Jufa Li performed the experiments, analyzed the data, authored or reviewed drafts of the article, and approved the final draft.

Jianfeng Shao conceived and designed the experiments, prepared figures and/or tables, authored or reviewed drafts of the article, and approved the final draft.

Ninghan Feng conceived and designed the experiments, prepared figures and/or tables, authored or reviewed drafts of the article, and approved the final draft.

Data Availability

The following information was supplied regarding data availability:

Raw data are available in the Supplemental Files.

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
