# Peer review of "Construction and verification of a prognostic model for bladder cancer based on disulfidptosis-related angiogenesis genes"

_PeerJ, doi:10.7717/peerj.18911_

## Round 0.1 · original submission · Major Revisions

· Academic Editor

Major Revisions

Two experts in the field have reviewed your manuscript. As you can see from their comments below, both raised many points about improving the manuscript. Please read them carefully and revise the manuscript accordingly. When you think some of these comments do not make sense, explain why. I am looking forward to your revised manuscript.

Reviewer 1 ·

Basic reporting

The manuscript introduces a prognostic model for bladder cancer (BLCA) by integrating disulfidptosis-related angiogenesis genes (ARGs) through bioinformatics analysis. The study leverages data from the TCGA and GEO databases to develop a risk model, validated through survival analysis and immune infiltration metrics. The model holds potential for clinical applications, aiding in risk assessment and therapeutic decisions in BLCA. While comprehensive, the manuscript would benefit from further refinement, clarification, and additional references to strengthen its scientific foundation.
Specific Comments
Abstract:
Line 17: The phrase “a freshly unveiled mode of cellular demise” could be rephrased as “a newly identified cell death mechanism” for conciseness.
Line 43: The word “upraised” is awkward; consider replacing it with “elevated” or “increased” for clarity.
Introduction:
Line 20:Provide general information for cancer treatment, cite this paper from NHS for more details: “Cancer treatments: Past, present, and future, 2024”
Line 25: Add recent statistics on BLCA incidence and mortality to underscore the clinical relevance of developing prognostic models.
Line 35: Provide a brief overview of disulfidptosis and its discovery in cancer research to establish context. Also, The paper should mention recent reviews in cell death pathways in general to introduce the topic into ferroptosis such as Targeting regulated cell death pathways in acute myeloid leukemia, 2023, orThe multidrug resistance transporter P-glycoprotein confers resistance to ferroptosis inducers, 2023 , other recently discovered cell death pathways should also be mentioned, such as cuproptosis (“Expression and potential immune involvement of cuproptosis in kidney renal clear cell carcinoma, 2023”,“Pan-cancer genetic analysis of cuproptosis and copper metabolism-related gene set, 2022”,“Pan-cancer profiles of the cuproptosis gene set, 2022”) and disulfidptosis (“Pan-cancer genetic analysis of disulfidptosis-related gene set, 2023”Actin cytoskeleton vulnerability to disulfide stress mediates disulfidptosis, 2023)
Line 65: Discuss previous attempts at BLCA prognostic models and the limitations that this study aims to address.
Materials and Methods:
Data Source and Processing (Section 2.1):
Line 80: Clarify how data quality and integrity from the TCGA and GEO databases were ensured, as this is crucial for reproducibility.
Line 100: Describe why certain genes were selected for analysis, especially with respect to their role in disulfidptosis and angiogenesis.
Unsupervised Clustering Analysis (Section 2.2):
Line 114: Expand on the rationale for using consensus clustering and how the optimal cluster number was determined.
Line 125: Explain why k=2 was chosen as the cutoff for clustering analysis, especially if there was consideration of other cluster numbers.
Differential and Enrichment Analysis (Section 2.3):
Line 140: Provide a brief background on the significance of GO and KEGG pathway analyses in understanding the functions of ARGs in cancer.
Line 160: Clarify why |log2FC| >1 and p <0.05 were chosen as thresholds, linking these choices to study objectives.
Prognostic Model Construction (Section 2.4):
Line 200: Briefly describe the advantages of LASSO regression in selecting prognostic genes, especially compared to alternative approaches like Cox regression.
Line 210: Explain the choice of ROC curve analysis for model validation and its relevance for clinical prediction accuracy.
Results:
Identification of Disulfidptosis-Related Subgroups (Section 3.1):
Line 240: Summarize key clinical differences between subgroups identified through clustering, as these differences may inform potential clinical applications.
Figure 2: Add annotations or labels to make distinctions between subgroups clearer, improving readability.
Differential and Functional Enrichment (Section 3.2):
Line 280: For the GO and KEGG analyses, elaborate on the biological significance of the enriched pathways (e.g., extracellular matrix regulation, immune cell chemotaxis) in BLCA.
Figure 3: Provide more descriptive figure captions, specifying the importance of each category (BP, CC, MF) for interpreting the results.
Prognostic Model Validation (Section 3.3):
Line 310: Emphasize the clinical significance of key genes (e.g., COL5A2, SCG2) and their roles in angiogenesis within BLCA.
Figure 4: Enhance the ROC and survival curve captions to include AUC values and clarify interpretations for readers.
Immune Microenvironment and Tumor Mutation Burden Analysis (Sections 3.5 and 3.6):
Line 360: Discuss the relevance of immune cell infiltration findings, specifically the implications of lower CD8+ T cells and dendritic cells in high-risk groups.
Figure 6: Explain the rationale behind using TIDE and Exclusion scores in TME analysis, emphasizing their predictive value for immunotherapy response.
Discussion:
Line 400: Provide a balanced view by discussing limitations of the model, such as reliance on retrospective datasets and potential biases introduced by cohort composition. Also discuss bias from the data, suggest to cite “Genetic expression in cancer research: Challenges and complexity, 2024” for more information.
Line 420: Compare this prognostic model with existing models in BLCA, highlighting any unique contributions or improvements.
Line 460: Emphasize the potential translational applications, such as using risk scores to tailor immunotherapy and combination treatments in clinical settings.
Conclusion:
Line 510: Summarize key findings and potential clinical applications, specifically for personalized treatment planning in BLCA.
Line 520: Suggest future research directions, such as validating the model in prospective cohorts and exploring the mechanisms of COL5A2 and SCG2 in BLCA progression.
With these revisions, the manuscript will provide a robust resource for understanding the prognostic potential of disulfidptosis-related ARGs in BLCA and inform future clinical applications.

Experimental design

see above

Validity of the findings

see above

Additional comments

see above

Reviewer 2 ·

Basic reporting

-

Experimental design

-

Validity of the findings

-

Additional comments

General comment
The manuscript entitled “Construction and Verification of a Prognostic Model for Bladder Cancer based on Disulfidptosis-Related Angiogenesis Genes” presents a comprehensive investigation into the development and validation of a prognostic risk model based on disulfidptosis-related genes (DRGs) for bladder cancer (BLCA). The study provides novel insights into the association between risk scores, clinical features, tumor microenvironment (TME), and immune characteristics, along with external validation using the GEO dataset. The integration of bioinformatics analyses and experimental validation strengthens the credibility of the findings. However, the manuscript would benefit from improvements in clarity, consistency, and addressing methodological limitations. In detail:
INTRODUCTION
Regarding the epidemiology and risk factors of BLCA also see: 10.23736/S2724-6051.22.04953-9
Expand on the uniqueness of disulfidptosis as a biological process and its relevance to cancer biology to emphasize its novelty.
Include a more detailed overview of why SCG2 and COL5A2 were chosen as focal points, as these genes seem central to the study.
MATERIALS AND METHODS
Provide more details about the unsupervised clustering analysis methodology, including the criteria for grouping samples.
Clarify the preprocessing steps for TCGA-BLCA data, including normalization and filtering criteria for differentially expressed genes.
RESULTS
Discuss why the AUC values for TCGA (0.613 and 0.624) are lower compared to the GEO validation (0.803 and 0.762). This discrepancy needs interpretation.
Enhance the explanation of scatter plots and heatmaps to clarify how high-risk individuals are characterized.
Highlight the implications of the lower mutation frequency observed in the high-risk cohort.
DISCUSSION
Elaborate on the potential mechanisms through which COL5A2 and SCG2 influence TME and tumor progression.
Acknowledge the limitations of using public datasets in greater detail, particularly the potential biases in patient demographics.
Propose specific steps for future research, such as validating the model in larger, ethnically diverse cohorts or exploring disulfidptosis pathways in greater detail.

---

## Round 0.2 · accepted · Accept

· Academic Editor

Accept

Since both reviewers now recommend accepting your revised manuscript, I am happy to inform you that I recommend its acceptance to the section editor. Congratulations!

Reviewer 1 ·

Basic reporting

good for publish

Experimental design

good for publish

Validity of the findings

good for publish

Additional comments

good for publish

Reviewer 2 ·

Basic reporting

The authors improved the manuscript according to previous suggestions. No further corrections are required.

Experimental design

The authors improved the manuscript according to previous suggestions. No further corrections are required.

Validity of the findings

The authors improved the manuscript according to previous suggestions. No further corrections are required.

Additional comments

The authors improved the manuscript according to previous suggestions. No further corrections are required.